# Differential Protective Effect of Zinc and Magnesium for the Hepatic and Renal Toxicity Induced by Acetaminophen and Potentiated with Ciprofloxacin in Rats

**DOI:** 10.3390/medicina60040611

**Published:** 2024-04-08

**Authors:** Alexandra Ciocan (Moraru), Diana Ciubotariu, Cristina Mihaela Ghiciuc, Mihnea Eudoxiu Hurmuzache, Cătălina Elena Lupușoru, Radu Crișan-Dabija

**Affiliations:** 1Faculty of Medicine, Department of Pharmacology, University of Medicine and Pharmacy “Grigore T. Popa”, 700115 Iaşi, Romania; morarualexandra89@yahoo.com (A.C.); cristina.ghiciuc@umfiasi.ro (C.M.G.); mihneaeudoxiu.hurmuzache@gmail.com (M.E.H.); elena.lupusoru@umfiasi.ro (C.E.L.); radu.dabija@umfiasi.ro (R.C.-D.); 2Clinical Hospital of Pulmonary Diseases, 400012 Iaşi, Romania; 3“St. Maria” Clinical Hospital for Children, 700309 Iaşi, Romania; 4“St. Parascheva” Hospital for Infectious Diseases, 700116 Iaşi, Romania

**Keywords:** ciprofloxacin, acetaminophen, hepatoprotection, zinc, magnesium

## Abstract

*Background and Objectives*: The purpose of this study was to investigate the influence induced by magnesium chloride (MgCl_2_) and zinc gluconate (ZnG) supplementation on liver and kidney injuries experimentally induced with acetaminophen (AAPh) and potentiated by a ciprofloxacin addition in rats. *Material and Methods*: The experiment was performed on five animal groups: group 1—control, treated for 6 weeks with normal saline, 1 mL/kg; group 2—AAPh, treated for 6 weeks with AAPh, 100 mg/kg/day; group 3—AAPh + C, treated for 6 weeks with AAPh 100 mg/kg/day and ciprofloxacin 50 mg/kg/day, only in the last 14 days of the experiment; group 4—AAPh + C + Mg, with the same treatment as group 3, but in the last 14 days, MgCl_2_ 10 mg/ kg/day was added; and group 5—AAPh + C + Zn, with the same treatment as group 3, but in the last 14 days, zinc gluconate (ZnG), 10 mg/kg/day was added. All administrations were performed by oral gavage. At the end of the experiment, the animals were sacrificed and blood samples were collected for biochemistry examinations. *Results*: Treatment with AAPh for 6 weeks determined an alteration of the liver function (increases in alanine aminotransferase, aspartate aminotransferase, lactic dehydrogenase, and gamma-glutamyl transferase) and of renal function (increases in serum urea and creatinine) (*p* < 0.001 group 2 vs. group 1 for all mentioned parameters). Furthermore, the antioxidant defense capacity was impaired in group 2 vs. group 1 (superoxide dismutase and glutathione peroxidase activity decreased in group 2 vs. group 1, at 0.001 < *p* < 0.01 and 0.01 < *p* < 0.05, respectively). The addition of ciprofloxacin, 50 mg/kg/day during the last 14 days, resulted in further increases in alkaline phosphatase, alanine aminotransferase, aspartate aminotransferase, urea, and creatinine (0.01 < *p* < 0.05, group 3 vs. group 2). MgCl_2_ provided a slight protection against the increase in liver enzymes, and a more pronounced protection against the increase in serum urea and creatinine (0.001 < *p* < 0.01 group 4 vs. group 3). MgCl_2_ provided a slight protection against the decrease in superoxide dismutase (0.01 < *p* < 0.05 group 4 vs. group 3), but not against decrease of glutathione peroxidase. The improvement of mentioned parameters could also be seen in the case of ZnG, to a higher extent, especially in the case of alanine aminotransferase and lactic dehydrogenase (0.01 < *p* < 0.05 group 5 vs. group 4). *Conclusions*: This study presents further proof for the beneficial effect of magnesium and zinc salts against toxicity induced by different agents, including antibacterials added to the analgesic and antipyretic acetaminophen; the protection is proven on the liver and kidney’s function, and the antioxidant profile improvement has a key role, especially in the case of zinc gluconate.

## 1. Introduction

Antibacterial drugs represent an important arsenal in fighting against infections. Among the greatest problems related to their utilization, bacterial resistance development is particularly severe, especially in the case of injudicious use; but another very serious concern related to antibacterial use is their potential to induce adverse effects, as their action in the patient’s body is not limited to the pathogenic microbial agent. Even though these drugs are regularly bio-transformed in the human body and eliminated, and their toxicity is generally selective, they often induce serious adverse effects or significant toxicity (nephrotoxicity, hepatotoxicity, cardiotoxicity) [1].

The toxicity of antibacterials often overlaps with that of some other drugs that they are concomitantly recommended with, or with that of recreational toxics (e.g., ethanol). Fever is a symptom that frequently accompanies infections, including bacterial ones. Acetaminophen is often used as an analgesic and antipyretic, and its association with antibacterials is common in the therapy of infections. It is considered safe at therapeutic levels, but at high doses it can lead to hepatotoxicity and nephrotoxicity. Although high doses of acetaminophen establish a conjugated bond with glucuronic acid or sulphate, a significant portion is metabolized by the cytochrome P450 system, determining the production of reactive toxic metabolites such as N-acetyl-p-benzoquinone imine. This interacts with sulfhydryl groups in the glutathione molecule, determining an impairment of antioxidant capacity and potentiating lipid peroxidation; several studies have reported that high acetaminophen doses reduce tubular epithelial cell vitality, leading to kidney toxicity [2,3,4]. The toxicity of acetaminophen is often cumulative with that of antibacterial medication.

On the other hand, certain chemical elements, either essential macronutrients or trace elements (chemicals representing less than a thousandth of the dry tissue composition in the human or animal body) have essential roles in the living world. These roles are often ignored, and the diet does not always manage to provide a necessary supply of certain such elements. The deficiency of both magnesium and zinc are relatively common [5,6,7,8] and known to be associated with kidney and liver function impairment [9,10,11,12,13,14,15,16,17,18,19]. For instance, several liver diseases, including cirrhosis, are associated with magnesium deficiency, and, in turn, insufficient magnesium aggravates those conditions [9,10]. Benefits of supplementation with zinc and magnesium have also been shown in the absence of deficiencies. Magnesium, for example, induces hepatoprotection against toxicity induced by arsenic trioxide [20], or elevated fructose levels [21], and clinical studies have demonstrated that an elevated magnesium level is associated with longer survival with chronic kidney disease [22]. Beneficial effects of zinc supplementation have been observed in children and adolescents with kidney disease [23], while a zinc-acetate formulation induced liver function improvement [24].

Based on the above, we justify the current study, in which we aimed to investigate the supposed protective effect of supplementation with magnesium chloride and zinc gluconate, respectively, in nephro-hepatopathy experimentally induced with acetaminophen and potentiated by the addition of ciprofloxacin. Hepatotoxicity was documented by the increase in liver enzymes, and creatinine and urea were markers for the kidney toxicity.

## 2. Materials and Method

### 2.1. Ethical Policies

This study complied with the European Guidelines for Human and Animal Rights, Directive 2010/63/EU, and the Romanian Law of Research no. 206/27.05.2004. The research was approved by the Research Ethics Committee of the Grigore T. Popa University of Medicine and Pharmacy in Iaşi, Romania (6 August 2019).

### 2.2. Substances

Acetaminophen (AAPh), powder; ciprofloxacin, neat; magnesium chloride (MgCl_2_), anhydrous; and zinc gluconate (ZnG), powder were purchased from Sigma Chemical Co. (St. Louis, MO, USA).

Substances were prepared as solutions.

### 2.3. Laboratory Animals

5-week-old male albino Wistar rats (nonconsanguineous; n = 50) were used in the experiments. The animals weighed 150 ± 10 g each at the beginning of the experiment and were purchased from the I. C. Cantacuzino National Institute of Research and Development for Microbiology and Immunology, Bucharest, Romania. The animals were individually housed in polypropylene cages according to European standards in the Laboratory for Pharmacology Research in a controlled environment (temperature 23 ± 2 °C, 50–60% relative humidity, central ventilation and artificial light–dark cycles of 12 h/12 h). The animals had free access to water and standard food, which were provided ad libitum.

The rats were acclimatized for 7 days prior to the beginning of the experimental study.

### 2.4. Groups and Design of the Experiment

The rats were randomly divided into 5 groups of 10, and the experiment was conducted for 42 days.

Group 1 (control, C): the animals were treated for 6 weeks with normal saline, 1 mL/kg.

Group 2 (acetaminophen-induced hepatopathy, AAPh): the rats were treated for 6 weeks with acetaminophen, 100 mg/kg/day.

Group 3 (acetaminophen-induced hepatopathy plus ciprofloxacin, AAPh + C): the rats were treated for 6 weeks with acetaminophen, 100 mg/kg/day, by oral gavage. In addition, ciprofloxacin, 50 mg/kg, was added in the last 14 days of the experiment (50 mg /kg/day).

Group 4 (acetaminophen-induced hepatopathy plus ciprofloxacin plus magnesium, AAPh + C + Mg): the rats were treated for 6 weeks with acetaminophen 100 mg/kg/day. In addition, in the last 2 weeks of the experiment, ciprofloxacin, 50 mg/kg/day, and magnesium chloride, 10 mg/kg/day, were added.

Group 5 (acetaminophen-induced hepatopathy plus ciprofloxacin plus zinc, AAPh + C + Zn): the rats were treated for 6 weeks with acetaminophen, 100 mg/kg/day. In addition, in the last 2 weeks of the experiment, ciprofloxacin, 50 mg/kg/day, and zinc gluconate, 10 mg/kg/day, were added.

All solutions/substances were administered by oral gavage, in a unique daily dose, at 7.30. The solutions for each of the pharmacologically active substances administered had the concentration calculated so that the volume of each administration was 1 mL of solution/kg.

### 2.5. Biochemical Determination

Blood samples were collected from the carotid artery of sacrificed animals and stored in standard biochemistry vacutainers. In order to limit supplementary suffering of the animals (required by the institution’s Ethics Committee), blood sampling was not performed before the application of different treatments or during different stages of the experiment.

The following parameters, related mainly to the hepatic function, were determined: alanine aminotransferase or glutamic-pyruvic transaminase (ALT/GTP), aspartate aminotransferase or glutamic oxaloacetic transaminase (AST/GOP), lactic dehydrogenase (LDH), and gamma-glutamyl transferase, γ-GT (GGT), alkaline phosphatase (ALP), and their values were expressed in international unites per liter (IU/L), with a precision of 1 IU/L (except for GGT, where the precision was 0.01 IU/L).

The following parameters related mainly to the kidney function were determined: urea and creatinine and their values were expressed in mg per deciliter (mg/dL), with a precision of 1 mg/dL.

Biochemical determinations were performed using kits supplied by BD Biosciences Co. (Heidelberg, Germany) and a VITROS^®^ 350 Chemistry System biochemical analyzer (Johnson & Johnson Ortho-Clinical Diagnostics) (colorimetric methods).

The following parameters related mainly to the antioxidant status were determined: superoxide dismutase 1 (SOD1) and Glutathione peroxidase 1 (GPX 1). The determinations were performed using enzyme-linked immunosorbent assay (ELISA) kits from Cloud-Clone Corp, and an optical density reader by Perkin EnVision. Values were expressed in mg per deciliter (pg/mL), with a precision of 1 pg/mL.

### 2.6. Statistical Data Analysis

Statistical analysis was performed using SPSS software version 20 for Windows (IBM, Chicago, IL, USA). Data were first checked for normality in each group and for each tested parameter using the Shapiro–Wilk test. As the data showed a normal distribution, the results were expressed in graphs and tables as average ± standard, and one-way ANOVA was used to test for differences among groups.

## 3. Results

### 3.1. Body Weight, Growth, Food, and Water Intake

During the acclimatization period and the 42 days thereafter, the body weight of the animals in all groups increased constantly regardless of their treatments. None of the rats died during the 6 weeks of different treatment applications.

### 3.2. Biochemical Determinations

Treatment with acetaminophen for 6 weeks determined an alteration of:liver function: increases in ALT, AST, LDH, GGT, and ALP (*p* < 0.001, group 2 vs. control group 1);renal function: increases in serum urea and serum creatinine (*p* < 0.001, group 2 vs. control group 1);antioxidant defense capacity (decrease in SOD and GPX activity, 0.001 < *p* < 0.01 and 0.01 < *p* < 0.05, respectively, group 2 vs. control group 1).

The addition of ciprofloxacin, 50 mg/kg/day, during the last 14 days of AAph administration resulted in a further increase of the following serum biochemical parameters: ALT, AST, serum creatinine, and urea (0.01 < *p* < 0.05, group 3 vs. group 2), while ALP was even more seriously increased (0.001 < *p* < 0.01, group 3 vs. group 2), and SOD was increased (0.01 < *p* < 0.05, group 3 vs. group 2). The serum concentrations of GGT, LDH, and GpX were not modified by the addition of ciprofloxacin.

The magnesium chloride oral treatment, 10 mg/kg/day, addition during the 14 days of ciprofloxacin administration on the background of acetaminophen-induced toxicity provided a certain degree of protection against the alteration of certain assessed biochemical parameters, as follows:pronounced protection against the increase in serum concentration of serum urea and creatinine (0.001 < *p* < 0.01, group 4 vs. group 3);slight, yet significant protection against the increase in the serum concentration of liver enzymes ALT, AST, LDH, GGT, and ALP, and against the decrease in the serum concentration of SOD (0.01 < *p* < 0.05, group 4 vs. group 3);insignificant protection against the decrease in the serum concentration of GPx.

A zinc gluconate oral treatment, 10 mg/kg/day, addition during the 14 days of ciprofloxacin administration on the background of acetaminophen-induced toxicity provided a certain degree of protection against the alteration of certain assessed biochemical parameters, as follows:particularly pronounced protection against the increase in the serum concentration of liver enzymes ALT and LDH and serum creatinine (*p* < 0.001, group 5 vs. group 3);pronounced protection against the increase in the serum concentration of liver enzyme GGT and serum urea (0.001 < *p* < 0.01, group 5 vs. group 3);slight, yet significant protection against the increase in the serum concentration of liver enzymes AST, ALP, and the decrease of SOD and GPx (0.001 < *p* < 0.01, group 5 vs. group 3).

Only for ALT and LDH, the improvement was superior in the case of zinc gluconate compared to magnesium chloride (0.01 < *p* < 0.05, group 4 vs. group 5), while the opposite situation was not recorded for any parameter.

The complete list of values for the assessed biochemical parameters in rats can be found in Table 1.

## 4. Discussions

In the current study, we have used a model of hepato-nephropathy induced with acetaminophen oral administration (100 mg/kg/day, 42 days) and potentiated with 50 mg/kg/day ciprofloxacin (orally, only in the last 14 days). The chosen dose of acetaminophen induced significant increase in liver enzymes, while nephrotoxicity, documented as urea and creatinine increases, even though statistically significant, was not as severe.

Furthermore, the addition of ciprofloxacin to acetaminophen mainly affected the liver enzymes’ concentration, while the increase in the biochemical parameters related to kidney function was less evident.

We have demonstrated that both zinc gluconate and magnesium chloride offered a considerable protection against renal, and mainly against the hepatic toxicity induced with acetaminophen and enhanced by ciprofloxacin in rats. The doses of zinc and magnesium compounds were chosen according to the medical literature, in order not to be toxic and yet to provide efficacy [25,26]. The choice of the oral route is justified by the fact that magnesium and zinc compounds are regularly given as dietary supplements, with the mentioned administration route.

The protection is more evident for the zinc-based preparation, especially in the case of liver function. Nephrotoxicity was less severe than hepatotoxicity in the case of the acetaminophen-only group 2. Additionally, in the ciprofloxacin + acetaminophen group 3, the ciprofloxacin addition just slightly potentiates the increase in urea and creatinine induced with acetaminophen; therefore, we consider that the results related to the protection induced with magnesium or zinc are difficult to be interpreted with relevance to animals or human beings.

Both compounds offered a slight level of protection against the alteration of anti-oxidant capacity induced by mentioned toxics (they protected against the decrease in the serum concentration of superoxide dismutase, but just ZnCl_2_ protects against the decrease in the serum concentration of glutathione peroxidase). This supports the hypothesis that magnesium or zinc cations kidney and liver protection is, at least partially, due to improving the antioxidant defense.

An important limitation of the study is the lack of modeling the infective pathology, as antibacterials (especially if used systemically, but also when used topically), and the pathology itself may contribute to changes in the parameters of the kinetics and dynamics of different other substances, as well as microelements or magnesium.

In the medical literature, there is important evidence from studies by different researchers regarding the possible beneficial effects of zinc and magnesium compounds in liver and kidney damage of various causes. There are both observational clinical studies that could prove a possible association of zinc or magnesium deficiency with liver or kidney pathology and interventional clinical studies documenting a possible benefit of the administration of zinc or magnesium-containing compounds in liver or kidney pathology). Furthermore, there are plenty of experimental studies proving the benefits of zinc or magnesium compounds in liver or kidney pathology, or the potentiation of liver or kidney damage by the deficiency of the two cations.

Some of those studies also manage to reveal certain aspects regarding mechanisms involved in hepatoprotection and nephroprotection induced by zinc and magnesium ions; in vitro studies contribute to the understanding of the protective mechanisms as well.

### 4.1. Magnesium—Roles in Hepatoprotection and Nephroprotection

#### 4.1.1. The Role of Magnesium in Hepatoprotection

The relationship between magnesium levels in the human body and liver diseases is complex. *Observational* studies showed that several liver diseases are associated with magnesium deficiency, while such a condition aggravates these diseases, like in a downward spiral [9]. Patients with liver cirrhosis usually have low magnesium body levels [10]. Cirrhosis is often due to alcohol consumption, and magnesium deficiency is common among alcohol-consumers [11]. Progression to cirrhosis is very likely, particularly in patients with low hepatocytic magnesium concentrations.

Hepatic loss of magnesium is associated with greater collagen deposition in the liver [27]. Decreased intracellular Mg^2+^ content has a negative impact on mitochondrial bioenergetics, which implies impairment of oxidation in hepatocytes, associated with reduction in ATP production and hepatocyte damage. The subsequent repair process of the liver leads to further fibrosis and worsens cirrhosis.

*Interventional* studies have shown that magnesium supplementation ameliorates various liver diseases. For instance, a 100 mg increase in daily magnesium intake is associated with a 49% decrease in the risk of mortality from all liver diseases [28]. Another study claims that magnesium supplementation can prevent hepatitis C virus replication by binding to the NS3 helicase of the virus [29].

The magnesium hepatoprotective effect has also been demonstrated by numerous *experimental studies*. Prophylactic administration of magnesium isoglycyrisinate (9 and 18 mg/kg/day) significantly reduced serum ALT and AST levels in rats treated with 20 mg/kg methotrexate (intravenously), and attenuated methotrexate-induced liver fibrosis, hepatocyte apoptosis, and reduced serum malondialdehyde levels more than glutathione, 80 mg/kg/day. Methotrexate-induced cyclooxygenase-2 expression, intestinal permeability, and inflammation were attenuated [30]. Magnesium isoglycyrrhizinate (i.p., 50 mg/kg/day) induces hepatoprotection against arsenic trioxide-induced toxicity in mice [20] or elevated fructose levels [21] in rats.

An in vitro study showed that magnesium cantharidate has an inhibitory effect on the proliferation of SMMC-7721 human hepatoma cells by blocking the MAPK (mitogen-activated protein kinases) signaling pathway; the phosphorylation levels of C-jun N-terminal kinase (JNK) and extracellular signal-related kinase (ERK) decrease significantly after such treatment [31].

#### 4.1.2. The Role of Magnesium in Nephroprotection

*Pathophysiological considerations*: The kidney has a vital role in the homeostasis of magnesium; renal Mg excretion is highly adaptive, but it is impaired when renal function declines significantly. In chronic kidney disease (CKD) of moderate severity, the increase in Mg^2+^ excretion largely compensates for the loss of the glomerular filtration rate, and normal serum magnesium levels are maintained. However, in more advanced forms of the disease, this compensatory mechanism becomes inadequate, so hypermagnesemia frequently develops in patients with creatinine clearance below 10 mL/min (as less Mg^2+^ is filtered through the glomeruli to be excreted). The treatment of hyperphosphatemia also contributes to hypermagnesemia [32]. In fact, CKD can also be associated with hypomagnesemia [12]. Both patients with polycystic kidney disease and end-stage renal disease on dialysis usually have normal serum Mg, and sometimes even hypomagnesemia.

*Human studies*: Observational studies showed that patients with polycystic kidney disease have severely depressed intestinal Mg^2+^ absorption, probably due to a deficiency of active vitamin D, and in end-stage renal disease the adaptive increase in active intestinal absorption of magnesium is impaired [13].

An interventional study showed that a higher level of proteinuria was associated with greater urinary Mg excretion. Mg oxide administration significantly increased serum Mg levels after one year, but only among patients with a urinary protein-to-creatinine ratio of less than 0.3 g/g, while there was no change in serum Mg in those with greater levels of proteinuria, suggesting that hypomagnesemia is a consequence of tubular injury [33].

*Experimental studies* demonstrated the protective effect of magnesium on the nephrotoxicity of colistin, decreasing serum urea and creatinine, but increasing the concentration of malondialdehyde, and improving renal histopathological aspects [34]. A low-magnesium diet exacerbated kidney damage induced by the high-phosphate diet. Increasing dietary magnesium may be helpful to attenuate phosphate-induced renal injury in mice [35].

Magnesium is known to protect against phosphate-induced tubular cell damage in vitro [36].

### 4.2. Zinc—Role in Hepatoprotection and Nephroprotection

Also, in the case of zinc, results of different studies support both the idea of zinc deficiency as an element associated with liver or kidney pathology, and the idea of the opportunity of dietary zinc supplementation in various forms of kidney pathology and hepatic disorders.

#### 4.2.1. The Role of Zinc in Hepatoprotection

The association of zinc deficiency with liver failure is clearly proven in experimental models and in clinical studies. The idea of dietary zinc supplementation appears to be supported by both observational and interventional clinical studies.

Since the 1950s, *observational studies* have demonstrated that serum Zn levels are associated with liver pathology. Vikbladh (1950; 1951) [14,15] reported that serum zinc was low in many chronic diseases, and in 1956, Bartholomay et al. [16] reported that serum Zn concentration was low in patients with liver cirrhosis and suggested that this was a consequence of hyperzincuria (increased urinary Zn excretion in such a condition being somewhat paradoxical).

Interventional studies have shown that zinc supplementation improves various liver conditions.

Some cirrhotic patients who experienced night blindness did not respond to vitamin A supplementation, but responded to zinc administration [37]. Zinc supplementation ameliorated liver fibrosis in patients with early cirrhosis [38], and supplementation with a zinc-acetate formulation significantly improved liver function [24]. Zinc therapy has been shown to be beneficial in subjects with hepatic encephalopathy [39].

Their results are reinforced by *experimental studies* that demonstrated the protective effect of zinc on liver damage induced by D-galactosamine or nickel in rats [40,41]. Pretreatment with zinc chloride (50 or 100 mg/kg daily, 36 days) reduced hepatic toxicity of glyphosate, ameliorated numerous biochemical parameters (such as increases in serum enzymes associated with hepatobiliary injury), but were ineffective in preventing histological lesions [42].

The ***mechanism*** underlying the association between zinc deficiency and liver dysfunction is insufficiently elucidated. These reports indicate an association between zinc deficiency and organ damage due to fibrosis and oxidative stress [43].

#### 4.2.2. The Role of Zinc in Nephroprotection

The association of zinc deficiency with renal failure, especially in hemodialyzed patients, is clearly proven, and so are the beneficial effects of zinc in various forms of renal pathology.

*Observational studies* have shown that serum zinc levels tend to decrease with the progression of CKD [17,18]. Survival analysis showed that zinc deficiency is a risk factor for progression to end-stage renal disease and death [19].

*Interventional studies* have shown that zinc supplementation improves the response to erythropoietin therapy in dialysis patients [44], and beneficial effects of zinc supplementation have been observed in children and adolescents with kidney disease [23]. A clinical trial demonstrated that zinc supplementation reduces urinary albumin excretion in type 2 diabetes patients with microalbuminuria [45]. In patients with low Zn levels, the risk of progression of primary kidney disease was lower if zinc-containing compounds were administered during the observation period [19].

*Experimental studies* demonstrated that in diabetic rats, zinc supplementation suppresses pathological changes associated with tubulointerstitial and glomerular damage [46].

Pretreatment with zinc chloride (50 or 100 mg/kg daily, 36 days) reduced renal and hepatic toxicity of glyphosate, and ameliorated numerous biochemical parameters (serum accumulation of creatinine), but did not alleviate histological damage in rats [42].

#### 4.2.3. Mechanisms Involved in Zinc Hepatoprotection and Nephroprotection

The mechanism underlying the association between zinc deficiency and renal dysfunction remains unclear. Various fundamental studies have shown that zinc is a regulator of oxidative stress, in its capacity as a cofactor of superoxide dismutase; its deficiency induces oxidative stress and kidney damage through nicotinamide adenine dinucleotide phosphate (NADPH) oxidase [47]. Evidence indicates that oxidative stress is the common denominator among major pathways involved in the development and progression of kidney diseases [48], with NADPH oxidase being identified as a major source of oxidative stress in kidney diseases. An intervention study in healthy individuals also demonstrated that zinc supplementation reduced oxidative stress [8]. Zinc deficiency promotes renal fibroblast activation and leads to interstitial fibrosis in diabetic mice [49].

Supplementation with zinc and/or magnesium should be recommended in patients undergoing long term antibiotherapy, combined with antipyretic administration. The idea is also sustained by the low price and low toxicity of such compounds, and by their beneficial effect on the immune system [5,8,50].

## 5. Conclusions

The hepatotoxicity of the antipyretic and analgesic drug acetaminophen is potentiated by ciprofloxacin administration in rats, but both zinc gluconate and magnesium chloride slightly alleviate the combined toxicity of the mentioned substances (diminish the elevated level of liver enzymes and creatinine). Especially in the case of the zinc compound, where the effect is slightly higher, this protection seems to partially associated with the potentiation of the antioxidant effect.

Administration of dietary supplements with magnesium and especially zinc may represent an easy, yet efficient way to reduce the toxicity of antibacterials associated with cyclo-oxygenase’s inhibitors.

## Figures and Tables

**Table 1 medicina-60-00611-t001:** The values for the assessed biochemical parameters in rats.

Assessed Biochemical Parameter/Group	Control	AAPh	AAPh + C	AAPh + C + Mg	AAPh + C + Zn
ALT (UI/L)	69.57 ± 2.57	2475.14 ± 284.38 *** vs. control	2964.57 ± 343.91 *** vs. control * vs. AAPh	2558.43 ± 251.97 *** vs. control NS vs. AAPh * vs. AAPh + C	2284.14 ± 205.55 *** vs. control NS vs. AAPh *** vs. AAPh + C * vs. AAPh + C + Mg
AST (UI/L)	154.43 ± 13.20	578.86 ± 61.56 *** vs. control	650.43 ± 59.33 *** vs. control * vs. AAPh	585.71 ± 28.61 *** vs. control NS vs. AAPh * vs. AAPh + C	583.57 ± 29.31 *** vs. control * vs. AAPh * vs. AAPh + C NS vs. AAPh + C + Mg
LDH (UI/L)	456.43 ± 117.74	3309 ± 287.64 *** vs. control	3463.86 ± 228.73 *** vs. control NS vs. AAPh	3139.57 ± 302.39 *** vs. control NS vs. AAPh * vs. AAPh + C	2806.14 ± 201.28 *** vs. control *** vs. AAPh *** vs. AAPh + C * vs. AAPh + C + Mg
GGT (× 100 UI/L)	33.57 ± 5.97	67.14 ± 6.12 *** vs. control	69.85 ± 5.73 *** vs. control NS vs. AAPh	60.86 ± 7.10 *** vs. control NS vs. AAPh * vs. AAPh + C	59.71 ± 4.79 *** vs. control * vs. AAPh ** vs. AAPh + C NS vs. AAPh + C + Mg
ALP (× 100 UI/L)	105 ± 6.66	166.57 ± 19.04 *** vs. control	205 ± 15.21 *** vs. control ** vs. AAPh	189.57 ± 9.24 *** vs. control * vs. AAPh * vs. AAPh + C	188.29 ± 12.75 *** vs. control * vs. AAPh * vs. AAPh + C NS vs. AAPh + C + Mg
Ureea (mg/dL)	31.43 ± 2.82	63.14 ± 6.15 *** vs. control	71.57 ± 7.00 * vs. AAPh	61.57 ± 4.72 *** vs. control NS vs. AAPh ** vs. AAPh + C	60.28 ± 4.31 *** vs. control * vs. AAPh ** vs. AAPh + C NS vs. AAPh + C + Mg
Creatinine (mg/dL)	5.91 ± 1.17	8.42 ± 0.80 *** vs. control	9.33 ± 0.59 *** vs. control * vs. AAPh	7.66 ± 1.11 * vs. control NS vs. AAPh ** vs. AAPh + C	7.19 ± 0.84 * vs. control * vs. AAPh ** vs. AAPh + C NS vs. AAPh + C + Mg
SOD (pg/mL)	24.71 ± 3.45	19.29 ± 1.50 ** vs. control	17.42 ± 1.27 *** vs. control * vs. AAPh	19.57 ± 1.90 ** vs. control NS vs. AAPh * vs. AAPh + C	20.43 ± 2.30 * vs. control * vs. AAPh * vs. AAPh + C NS vs. AAPh + C + Mg
GPX (pg/mL)	31.57 ± 3.60	26 ± 4.08 * vs. control	23,42 ± 3.46 ** vs. control NS vs. AAPh	26.43 ± 5.71 NS vs. control NS vs. AAPh NS vs. AAPh + C	27.71 ± 3.77 NS vs. control NS vs. AAPh * vs. AAPh + C NS vs. AAPh + C + Mg

Legend: NS—statistically insignificant; *—0.01 < *p* < 0.05; **—0.001 < *p* < 0.01; ***—*p* < 0.001.

## Data Availability

Data are contained within the article.

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
