# Peer review of "Differential Protective Effect of Zinc and Magnesium for the Hepatic and Renal Toxicity Induced by Acetaminophen and Potentiated with Ciprofloxacin in Rats"

_medicina, 2024, doi:10.3390/medicina60040611_

Round 1
Reviewer 1 Report
Comments and Suggestions for Authors
Introduction can be improved by several points: (1) Describing more clearly the hepato- and nephro- protective impact of Zn or Mg in presence or absence of Zn or Mg deficiency. Now the text in the introduction and more explicitly in the discussion part mix everything into one basket. The design of study is however relevant to the situation without deficiencies. (2) The selected dosages of Mg and Zn should be more justified by literature data. (3) Authors mention liver function, although test liver injury markers only. This discrepancy should be justified. (4) There is necessary to have clear justification to use acetaminophen as nephrotoxicant.
The design of the study is described clearly enough, although the comments mentioned for introduction questions appropriateness of study design chosen.
Methods. The methods are clear but the comments mentioned for introduction questions appropriateness of methodology chosen.
Results. The results are presented clearly, but (1) there seems to be inappropriate values presented in the 5th animal group for urea and creatinine measurements. (2) It is assumed that all animals survived until sacrifice. This should be clearly stated. (3) baseline values are missing. This missing information should be corrected or justified.
Conclusions. (1) Conclusions and discussion before are provided without critical appraisal of functional relevance of the results. (2) Although hepatotoxicity seems as clearly proven in 2nd and 3rd group (although not necessary with functional consequences), the nephrotoxicity is very small and statistically significant improvements in urea and creatinine are of very questionable relevance to animals and humans. (3) Discussion part should be more focused on remarks given before and substantially shortened.
Author Response
Thank you for your valuable comments.
Those comments made us aware of the correction necessary for the value of creatinine serum concentration (a total mistake, explained below).
Also, we have shortened the “Discussion” section, making it more focused on own results.
We have substantially changed “Introduction” section and explained why periodic or baseline blood sampling was not performed (to reduce animals’ suffering, a condition imposed by the Ethics Committee).
Introduction can be improved by several points:
(1) Describing more clearly the hepato- and nephro- protective impact of Zn or Mg in presence or absence of Zn or Mg deficiency. Now the text in the introduction and more explicitly in the discussion part mix everything into one basket. The design of study is however relevant to the situation without deficiencies.
Now, the 4th paragraph in the Introduction section looks like below.
On the other hand, certain chemical elements, either essential macronutrients or trace elements (chemicals representing less than a thousandth of the dry tissue composition in the human or animal body) have essential roles in the living world. These roles are often ignored, and the diet does not always manage to provide a necessary supply of certain such elements. The deficiency of both magnesium and zinc are relatively common [5, 6, 7, 8] and known to be associated with kidney and liver function impairment [9, 10, 11, 12, 13, 14, 15, 16, 17, 18, 19]. For instance, several liver diseases, including cirrhosis, are associated with magnesium deficiency, and insufficient magnesium aggravates, in turn, those conditions [9, 10]. Benefits of supplementation with zinc and magnesium have been shown also in the absence of deficiencies. Magnesium, for example, induces hepatoprotection against toxicity induced by arsenic trioxide [20] or elevated fructose levels [21], and clinical studies have demonstrated that elevated magnesium level of it is associated with longer survival with chronic kidney disease [22]. Beneficial effects of zinc supplementation have been observed in children and adolescents with kidney disease [23], while a zinc-acetate formulation induced liver function improvement [24].
(2) The selected dosages of Mg and Zn should be more justified by literature data.
Such justification now appears in the “Discussion” section.
The doses of zinc and magnesium compounds were chosen according to medical literature, in order not to be toxic and yet to provide efficacy [25, 26]. The choice of the oral route is justified by the fact that magnesium and zinc compounds are regularly given as dietary supplements, with mentioned administration route.
Bandyopadhyay, B.; Bandyopadhyay, S. K. Protective Effect of Zinc Gluconate on Chemically Induced Gastric Ulcer. Indian J Med Res 1997, 106, 27–32.
Mohammed, K. A.; Goji, A. D. T.; Tanko, Y.; Muhammed, A.; Salisu, I. A. Protective Effects of Magnesium Chloride on Liver Enzymes and Biomarkers of Oxidative Stress in High Fat Diet Fed Rats. Niger J Physiol Sci 2019, 34 (2), 149–157.
(3) Authors mention liver function, although test liver injury markers only. This discrepancy should be justified.
Now, the 4th paragraph in the Introduction section looks like below.
Based on the above, we justify the current study, in which we aimed to investigate the supposed protective effect of supplementation with magnesium chloride, respectively zinc gluconate in nephro-hepatopathy experimentally induced with acetaminophen and potentiated by the addition of ciprofloxacin. Hepatotoxicity was documented by the increase in liver enzymes, and creatinine and urea were markers for the kidney toxicity.
(4) There is necessary to have clear justification to use acetaminophen as nephrotoxicant.
We have added few things to the introduction. Now, the second paragraph in the “Introduction” section looks like below.
The toxicity of antibacterials often overlaps with that of some other drugs that they are concomitantly recommended with or with that of recreational toxics (eg, ethanol). Fe-ver is a symptom that frequently accompanies infections, including bacterial ones. Acet-aminophen is commonly used as analgesic and antipyretic and its association with anti-bacterials is common in the therapy of infections; it is considered safe at therapeutic levels, but at high doses it can lead to hepatotoxicity and nephrotoxicity. Although high doses of acetaminophen establish a conjugated bond with glucuronic acid or sulphate, a signifi-cant portion is metabolized by the cytochrome P450 system, determining the production of reactive toxic metabolites such as N-acetyl-p-benzoquinone imine. This interacts with sulfhydryl groups in the glutathione molecule, determining an impairment of antioxidant capacity and potentiating lipid peroxidation; several studies have reported that high ac-etaminophen doses reduce tubular epithelial cell vitality, leading to kidney toxicity [2, 3, 4]. The toxicity of acetaminophen is often cumulative with that of antibacterial medication.
Kandemir, F.; Kucukler, S.; Eldutar, E.; Caglayan, C.; Gülçin, İ. Chrysin Protects Rat Kidney from Paracetamol-Induced Oxidative Stress, Inflammation, Apoptosis, and Autophagy: A Multi-Biomarker Approach. Sci. Pharm. 2017, 85 (1), 4. https://doi.org/10.3390/scipharm85010004.
Ucar, F.; Taslipinar, M. Y.; Alp, B. F.; Aydin, I.; Aydin, F. N.; Agilli, M.; Toygar, M.; Ozkan, E.; Macit, E.; Oztosun, M.; Cayci, T.; Ozcan, A. The Effects of N-Acetylcysteine and Ozone Therapy on Oxidative Stress and Inflammation in Acetaminophen-Induced Nephrotoxicity Model. Renal Failure 2013, 35 (5), 640–647. https://doi.org/10.3109/0886022X.2013.780530.
Yu, Y.-L.; Yiang, G.-T.; Chou, P.-L.; Tseng, H.-H.; Wu, T.-K.; Hung, Y.-T.; Lin, P.-S.; Lin, S.-Y.; Liu, H.-C.; Chang, W.-J.; Wei, C.-W. Dual Role of Acetaminophen in Promoting Hepatoma Cell Apoptosis and Kidney Fibroblast Proliferation. Molecular Medicine Reports 2014, 9 (6), 2077–2084. https://doi.org/10.3892/mmr.2014.2085.
The design of the study is described clearly enough, although the comments mentioned for introduction questions appropriateness of study design chosen.
Response: -----
Methods. The methods are clear but the comments mentioned for introduction questions appropriateness of methodology chosen.
Response: -----
Results. The results are presented clearly, but (1) there seems to be inappropriate values presented in the 5th animal group for urea and creatinine measurements. (2) It is assumed that all animals survived until sacrifice. This should be clearly stated. (3) baseline values are missing. This missing information should be corrected or justified.
(1) The true value for creatinine should be 7.19 ± 0.84, instead of 60.28 ± 4.31, which is the value for urea written twice.
(2) Now, in the text there is mentioned that no animal died during the 6 weeks of different treatments application. (in green, in the beginning of “Results” section) (None of the rats died during the 6 weeks of different treatments application.)
(3) The Institution Ethics Committee requires that the animals involved in the experiment should not be exposed to unnecessary suffering. So, we have considered to compare the control group biochemical values after 6 six with normal saline treatment with the values in groups with different treatments in order to limit the suffering due to obtaining the blood samples. This is mentioned in green in the section “Biochemical determination” (In order to limit supplementary suffering of the animals (required by the institution’s Ethics Committee), blood sampling was not performed before the application of different treatments or during different stages of the experiment.).
Conclusions.
(1) Conclusions and discussion before are provided without critical appraisal of functional relevance of the results.
(2) Although hepatotoxicity seems as clearly proven in 2nd and 3rd group (although not necessary with functional consequences), the nephrotoxicity is very small and statistically significant improvements in urea and creatinine are of very questionable relevance to animals and humans.
(1) and (2) Now, in the beginning of “Discussion” section we have introduced some new text. The paragraphs highlighted in green are answering specifically the requests in the comments.
In the current study, we have used a model of hepato-nephropathy induced with acetaminophen oral administration (100 mg / kg / day, 42 days) and potentiated with 50 mg / kg / day ciprofloxacin (orally, only in the last 14 days). The chosen dose of acetaminophen induced significant increase in liver enzymes, while nephrotoxicity, documented as urea and creatinine increase, even though statistically significant, was not as severe.
Also, the addition of ciprofloxacin to acetaminophen mainly affected the liver enzymes concentration, while the increase in the biochemical parameters related to kidney function was less evident.
We have demonstrated that both zinc gluconate and magnesium chloride offered a considerable protection against renal and mainly against the hepatic toxicity induced with acetaminophen and enhanced by ciprofloxacin in rats. The doses of zinc and magnesium compounds were chosen according to medical literature, in order not to be toxic and yet to provide efficacy [25, 26]. The choice of the oral route is justified by the fact that magnesium and zinc compounds are regularly given as dietary supplements, with mentioned administration route.
The protection is more evident for the zinc-based preparation, especially in the case of liver function. Nephrotoxicity was less severe than hepatotoxicity in case of acetaminophen-only group 2. Also, in ciprofloxacin + acetaminophen group 3, ciprofloxacin addition just slightly potentiates the increase in urea and creatinine induced with acetaminophen; therefore, we consider that the results related to the protection induced with magnesium or zinc are difficult to be interpreted with relevance to animals or human beings.
Both compounds offered a slight level of protection against the alteration of anti-oxidant capacity induced by mentioned toxics (they protected against the decrease in the serum concentration of superoxide dismutase, but just ZnCl2 protects against the decrease in the serum concentration of glutathione peroxidase). This supports the hypothesis that magnesium or zinc cations kidney and liver protection is, at least partially, due to improving the antioxidant defense.
…
Our study is particularly relevant for the situations when supplementation occurs in the absence of cations deficiency.
(3) Discussion part should be more focused on remarks given before and substantially shortened.
The “Discussion” section has been substantially changed.

Reviewer 2 Report
Comments and Suggestions for Authors
Below the table you need to place notes with a breakdown of the groups and indicators. It is necessary to explain the reason for the significant increase in creatinine concentration in the group with combined use of zinc and magnesium, since the opposite reaction is expected. It is not clear from the text and design of the study why the authors abandoned periodic blood sampling to study biochemical parameters, but took only one point. I consider the limitation of the study to be the fact that the experiment did not involve modeling the infection itself, since antibiotics in real practice are prescribed precisely because of local or generalized infection, which also contributes to changes in the parameters of the kinetics and dynamics of drugs, as well as microelements. The discussion section does not involve an actual discussion of your data, but is presented as an additional review of the literature and can be shortened. It is not clear why the authors, given the availability of experimental material, did not study the histology and pathomorphology of the liver and kidneys.
Comments on the Quality of English LanguageBelow the table you need to place notes with a breakdown of the groups and indicators. It is necessary to explain the reason for the significant increase in creatinine concentration in the group with combined use of zinc and magnesium, since the opposite reaction is expected. It is not clear from the text and design of the study why the authors abandoned periodic blood sampling to study biochemical parameters, but took only one point. I consider the limitation of the study to be the fact that the experiment did not involve modeling the infection itself, since antibiotics in real practice are prescribed precisely because of local or generalized infection, which also contributes to changes in the parameters of the kinetics and dynamics of drugs, as well as microelements. The discussion section does not involve an actual discussion of your data, but is presented as an additional review of the literature and can be shortened. It is not clear why the authors, given the availability of experimental material, did not study the histology and pathomorphology of the liver and kidneys.
Author Response
Thank you for your valuable comments.
Those comments made us aware of the correction necessary for the value of creatinine serum concentration (a total mistake, explained below).
Also, we have shortened the “Discussion” section, making it more focused on own results.
We have substantially changed “Introduction” section and explained why periodic or baseline blood sampling was not performed (to reduce animals’ suffering, a condition imposed by the Ethics Committee).
Below the table you need to place notes with a breakdown of the groups and indicators.
Before the table, now there is a legend introduced and the table has become easier to follow.
It is necessary to explain the reason for the significant increase in creatinine concentration in the group with combined use of zinc and magnesium, since the opposite reaction is expected.
The true value for creatinine should be 7.19 ± 0.84, instead of 60.28 ± 4.31, which is the value for urea written twice.
It is not clear from the text and design of the study why the authors abandoned periodic blood sampling to study biochemical parameters, but took only one point.
The Institution Ethics Committee requires that the animals involved in the experiment should not be exposed to unnecessary suffering. So, we have considered to compare the control group biochemical values after 6 six with normal saline treatment with the values in groups with different treatments in order to limit the suffering due to obtaining the blood samples. This is mentioned in green in the section “Biochemical determination” (In order to limit supplementary suffering of the animals (required by the institution’s Ethics Committee), blood sampling was not performed before the application of different treatments or during different stages of the experiment.).
I consider the limitation of the study to be the fact that the experiment did not involve modeling the infection itself, since antibiotics in real practice are prescribed precisely because of local or generalized infection, which also contributes to changes in the parameters of the kinetics and dynamics of drugs, as well as microelements.
The “Discussion” section now includes a paragraph assuming that idea.
An important limitation of the study is the lack of modeling the infective pathology, as antibacterials (especially if used systemically, but also when used topically) and the pathology itself may contribute to changes in the parameters of the kinetics and dynamics of other, as well as microelements or magnesium.
The discussion section does not involve an actual discussion of your data, but is presented as an additional review of the literature and can be shortened.
The “Discussion” section has been substantially changed.
It is not clear why the authors, given the availability of experimental material, did not study the histology and pathomorphology of the liver and kidneys.
We could not study the pathomorphology of the liver and kidneys due to the limited funds allocated for the PhD research.

Round 2
Reviewer 1 Report
Comments and Suggestions for Authors
No further comments
Reviewer 2 Report
Comments and Suggestions for Authors
No